# RadVLM-GRPO: Enhancing Chest X-ray Report Generation and Visual Grounding via Reinforcement Learning

**Benjamin Gundersen**[1][*]                                                 BENJAMIN.GUNDERSEN@UZH.CH

**Nicolas Deperrois**[1][*]                                                 NICOLAS.DEPERROIS@UZH.CH

**Samuel Ruiperez-Campillo**[2]                            SAMUEL.RUIPEREZCAMPILLO@INF.ETHZ.CH

**Thomas M. Sutter**[2]                                       THOMAS.SUTTER@INF.ETHZ.CH

**Julia E. Vogt**[2]                                               JULIA.VOGT@INF.ETHZ.CH

**Michael Moor**[3]                                             MICHAEL.MOOR@BSSE.ETHZ.CH

**Farhad Nooralahzadeh**[1,4][†]                          FARHAD.NOORALAHZADEH@UZH.CH

**Michael Krauthammer**[1][†]                              MICHAEL.KRAUTHAMMER@UZH.CH

[1] *Department of Quantitative Biomedicine, University of Zurich, Zurich, Switzerland*

[2] *Department of Computer Science, ETH Zurich, Zurich, Switzerland*

[3] *Department of Biosystems Science and Engineering, ETH Zurich, Basel, Switzerland*

[4] *Institute of Computer Science, Zurich University of Applied Sciences, Zurich, Switzerland*

**Editors:** Accepted for publication at MIDL 2026

## Abstract

Recent advances in vision-language models (VLMs) have improved Chest X-ray (CXR) interpretation in multiple aspects. However, many medical VLMs rely solely on supervised fine-tuning (SFT), which optimizes next-token prediction without evaluating answer quality. In contrast, reinforcement learning (RL) can incorporate task-specific feedback, and its combination with explicit intermediate reasoning ("thinking") has demonstrated substantial gains on verifiable math and coding tasks. To investigate the effects of RL and thinking in a CXR VLM, we perform large-scale SFT on CXR data to build an updated RadVLM based on Qwen3-VL, followed by a cold-start SFT stage that equips the model with basic thinking ability. We then apply Group Relative Policy Optimization (GRPO) with clinically grounded, task-specific rewards for report generation and visual grounding, and run matched RL experiments on both domain-specific and general-domain Qwen3-VL variants, with and without thinking. Across these settings, we find that while strong SFT remains crucial for high base performance, RL provides additional gains on both tasks, whereas explicit thinking does not appear to further improve results. Under a unified evaluation pipeline, the RL-optimized RadVLM models outperform their baseline counterparts and reach state-of-the-art performance on both report generation and grounding, highlighting clinically aligned RL as a powerful complement to SFT for medical VLMs[1].

**Keywords:** Vision Language Models, Group Relative Policy Optimization, Reinforcement Learning, Radiology, Chest X-ray

---

[*] Contributed equally.

[†] Shared senior authorship.

1. Code is available at https://github.com/uzh-dqbm-cmi/RadVLM-GRPO and the updated SFT and RL models will be released under a new version at https://physionet.org/content/radvlm-model

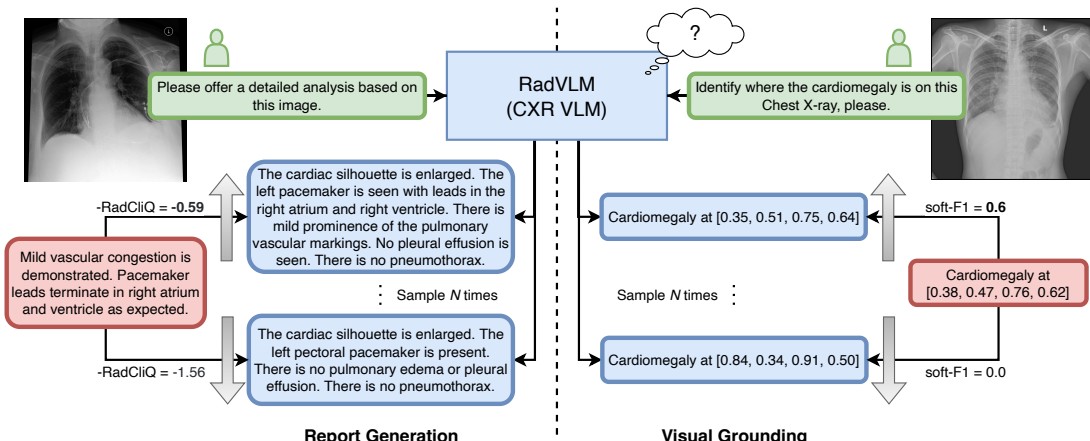

Figure 1: **Optimization of RadVLM with GRPO.** We optimize RadVLM with GRPO by sampling $N$ outputs from the model conditioned on an image-instruction pair. We then compare the outputs to the ground truth with a task-specific reward: RadCliQ for report generation and IoU-based metric (soft-F1) for visual grounding. These rewards provide an update signal to optimize RadVLM.

## 1. Introduction

Reinforcement learning has emerged as a central paradigm in modern reasoning-focused foundation models (Guo et al., 2025; Zheng et al., 2025a). Recent work in large language models (LLMs) has shown that RL, combined with explicit intermediate reasoning ("thinking"), can substantially improve performance on math and coding tasks (Shao et al., 2024; Guo et al., 2025; Yang et al., 2025; Yu et al., 2025). Rather than solely relying on supervised fine-tuning (SFT), these methods employ RL algorithms such as Group Relative Policy Optimization (GRPO) (Shao et al., 2024) to reward outputs. However, current medical VLMs specialized in CXR interpretation are typically optimized via image-instruction SFT-only (Chen et al., 2024; Sellergren et al., 2025; Deperrois et al., 2025b). Recent work applied GRPO in the medical and CXR domain on closed-ended tasks, such as visual question answering (VQA) and classification with predefined labels (Lai et al., 2025; Pan et al., 2025; Fan et al., 2025).

In contrast, the workflow of radiologists centers on producing open-ended, free-text reports, which could in principle be supported by VLMs trained directly on report generation (Tanno et al., 2025). However, the SFT objective – next-token prediction – is agnostic to the clinical utility of the generated report: it optimizes local likelihood rather than sequence-level, clinically meaningful criteria that RL could potentially optimize. For visual grounding, which plays a crucial role for radiology assistance (Lee et al., 2022), a similar limitation arises: token-level SFT-only matches coordinate strings, while the true objective is a geometric overlap between predicted and reference bounding boxes. These gaps motivate moving beyond SFT-only optimization and systematically studying post-training RL for report generation and visual grounding.

In that regard, previous work has explored RL for CXR report generation through offline direct preference optimization (DPO) (Hein et al., 2024; Liang et al., 2025), and more recently through online GRPO (Lin et al., 2025). Yet, the benefits of task-aligned RL for open-ended reporting and grounding remain insufficiently understood. In particular, it is unclear (i) how to best design rewards that jointly reflect lexical similarity and clinical correctness for reports, and continuous overlap-based feedback for grounding; (ii) whether performance gains arise from the RL objective alone or from its combination with preceding thinking; and (iii) to what extent SFT on domain-specific data is necessary when performing RL on that same domain.

To address these questions, we build an updated version of the state-of-the-art multitask conversational CXR-VLM RadVLM (Deperrois et al., 2025b), based on Qwen3-VL (Bai et al., 2025a) and focus on its GRPO-based optimization for report generation and visual grounding, as visualized in Figure 1. For report generation, we employ a clinically grounded reward, RadCliQ (Yu et al., 2023), while for visual grounding, we design a continuous IoU-based reward. We further synthesize thinking data, thereby equipping RadVLM with explicit thinking and non-thinking variants, and optimize both with GRPO. In addition, we apply the same RL procedure to out-of-domain Qwen3-VL-8B Instruct and Thinking models to assess to what extent a general-domain VLM can be adapted to the CXR domain by using RL alone. Under a unified evaluation pipeline, we find that training RadVLM with GRPO for grounding and report generation consistently improves all metrics over its SFT counterpart and outperforms state-of-the-art baselines on most metrics, while explicit thinking yields at best marginal additional gains over direct-answer variants.

In summary, the contributions of this work are as follows:

- We analyze large-scale post-training RL for report generation and visual grounding, differentiating from prior work focused on SFT or RL on closed-ended tasks.
- Through ablation studies, we reveal the impact of different components (SFT, RL, thinking) on model performance.
- We demonstrate that, in our setting, RL-optimized direct-answer models surprisingly match or outperform their thinking-augmented variants.
- We introduce a continuous soft-F1 reward for visual grounding and benchmark multiple reward formulations for report generation.
- Finally, we release our complete codebase and hyperparameters for SFT and GRPO to facilitate full reproducibility of high-performing CXR models.

## 2. Related Work

### 2.1. RL and thinking for LLMs

Recent advances in LLM research show that RL can markedly improve performance, especially when combined with explicit thinking and verifiable rewards. The OpenAI thinking series (OpenAI, 2024), open-weight models such as DeepSeek-R1 (Liu et al., 2024; Guo et al., 2025), and Qwen3 (Yang et al., 2025) typically combine a cold-start SFT stage, which teaches the model how to explicitly reason before providing an answer, followed by online RL. These RL-with-verifiable-reward (RLVR) pipelines yield large gains on tasks such as math and code, particularly when encouraging chain-of-thought reasoning before providing the final answers (Guo et al., 2025; Muennighoff et al., 2025; Hu et al., 2025).

However, their success strongly depends on reliable reward signals; for open-ended tasks, reward design is more challenging, and improvements are less consistent between tasks (Cheng et al., 2025).

## 2.2. RL and thinking for VLMs

Inspired by text-based results, the combination of RL and thinking has also been explored for VLMs, with a strong emphasis on visual-math tasks. Vision-R1 (Huang et al., 2025) synthesizes multimodal chain-of-thought data and applies GRPO to Qwen2.5-VL (Bai et al., 2025b), while VL-Rethinker (Wang et al., 2025) augments training with replay and "rethinking" triggers to improve visual-math. Beyond math, online RL also helps perception and grounding: Visual-RFT (Liu et al., 2025) and VLM-R1 (Shen et al., 2025) report improved localization and visual understanding over SFT with limited additional data. Still, it remains unclear how much benefit currently comes from explicit thinking versus reward shaping and exploration across different domains. Experiments on VLMs of similar size as RadVLM indicate that, for some tasks, applying GRPO to models with explicit chain-of-thought can even underperform models that directly produce the final answer (Li et al., 2025a; Lai et al., 2025). Moreover, Qwen3-VL (Bai et al., 2025a) is obtained through multi-stage training (including RL), and its instruct (non-thinking) variant outperforms the thinking variant on some tasks, including Document Understanding, 2D / 3D Grounding, Perception with Tools, and Multi-Modal Coding (see Appendix Table 10).

## 2.3. VLMs for radiology

Many CXR VLMs are trained via supervised instruction tuning on multi-task radiology data. General medical models such as LLaVA-Med (Li et al., 2023), MedGemma (Sellergren et al., 2025), as well as CXR-focused systems such as CheXagent (Chen et al., 2024), MAIRA-2 (Bannur et al., 2024), and RadVLM (Deperrois et al., 2025b), rely on a large instruction corpora covering report generation, classification, and grounding (Deperrois et al., 2025a). However, these models mostly rely on SFT without reinforcement learning integration.

Med-R1 (Lai et al., 2025), MedVLM-R1 (Pan et al., 2025), and ChestX-Reasoner (Fan et al., 2025) apply GRPO-style training to close-ended CXR/medical VQA, demonstrating gains in diagnostic accuracy and structured reasoning. For open-ended report generation, RL has so far predominantly been used in the form of offline direct preference optimization (DPO): CheXalign (Hein et al., 2024) and CheXPO (Liang et al., 2025) automatically construct preference pairs but do not study online RL or explicit thinking. DeepMedix-R1 (Lin et al., 2025) is the closest to our setting, applying online GRPO with thinking to report generation; however, it optimizes a combination of lexical rewards without incorporating an explicit clinical signal, and does not disentangle gains due to GRPO versus explicit chain-of-thought. Moreover, it incorporates bounding boxes as intermediate steps within thinking traces without taking correctness into account, while we treat visual grounding as a distinct downstream task with a specialized reward.

## 3. Methods

### 3.1. Preliminaries

#### 3.1.1. SUPERVISED FINE-TUNING

During fine-tuning on image-question-answer triples $(v, q, a)$, the model is trained to predict the current assistant tokens $x_{a,i}$ conditioned on visual tokens $x_v$, question tokens $x_q$ and previously generated assistant tokens $x_{a,<i}$. The supervised fine-tuning objective minimizes the negative log-likelihood: $\mathcal{L}_{\text{SFT}} = -\sum_{i=1}^{L_a} \log p(x_{a,i} \mid x_v, x_q, x_{a,<i})$.

#### 3.1.2. GROUP RELATIVE POLICY OPTIMIZATION

For a question-answer pair $(q, a)$, $G$ responses $\{o_i\}_{i=1}^{G}$ are sampled from the old policy $\pi_{\theta_{\text{old}}}$. The GRPO objective (Shao et al., 2024; Guo et al., 2025) is defined as:

$$
\mathcal{L}_{\text{GRPO}}(\theta) = \mathbb{E}_{(q,a)\sim\mathcal{D}, \{o_i\}_{i=1}^{G}\sim\pi_{\theta_{\text{old}}}(\cdot|q)}
$$
$$
\left[ \frac{1}{G} \sum_{i=1}^{G} \frac{1}{|o_i|} \sum_{t=1}^{|o_i|} \Big( \min\big(r_{i,t}(\theta)\,\hat{A}_{i,t}, \text{clip}(r_{i,t}(\theta), 1-\epsilon, 1+\epsilon)\,\hat{A}_{i,t}\big) - \beta\, D_{\text{KL}}(\pi_\theta\|\pi_{\text{ref}}) \Big) \right], \tag{1}
$$

with $r_{i,t}(\theta) = \frac{\pi_\theta(o_{i,t}|q,o_{i,<t})}{\pi_{\theta_{\text{old}}}(o_{i,t}|q,o_{i,<t})}$ and group advantage $\hat{A}_{i,t} = \frac{R_i - \text{mean}(\{R_i\}_{i=1}^{G})}{\text{std}(\{R_i\}_{i=1}^{G})}$.

Here, we prioritize an online reinforcement learning approach to enable dynamic exploration, avoiding the limitations of offline methods like DPO (Rafailov et al., 2023), used in Hein et al. (2024), which rely on fixed, static pairwise preferences. Among online algorithms, we select GRPO over PPO for its computational efficiency and demonstrated success in reasoning tasks (Guo et al., 2025).

### 3.2. Training data

We use two types of datasets: an instruction dataset and a cold-start dataset. The instruction dataset is the RadVLM dataset Deperrois et al. (2025b), a multimodal corpus of over one million image-text pairs. Each sample consists of a single frontal CXR image paired with a user-assistant interaction. The dataset spans four complementary instruction types: (i) free-text report generation (37%), where the assistant produces the findings section from a single image; (ii) abnormality classification (38%); (iii) visual grounding with bounding-box supervision, covering anatomical, abnormality, and phrase-level localization (15%); and (iv) multi-turn conversations generated with GPT-4o (9%). These components jointly support the training of models with fine-grained visual understanding and robust conversational behavior in radiology. Full details on the dataset construction and preprocessing are provided in Deperrois et al. (2025a,b).

In addition, we create a cold-start dataset consisting of thinking examples. To this end, we randomly sampled datapoints from the RadVLM instruction dataset, restricting our selection exclusively to report generation and visual grounding tasks. We then prompt Qwen3-VL-235B-Instruct (Bai et al., 2025a) to generate a thinking trajectory given the image and ground truth for both report generation and grounding (see Appendix I for prompt details). We then separate the thinking part from the final answer using the think

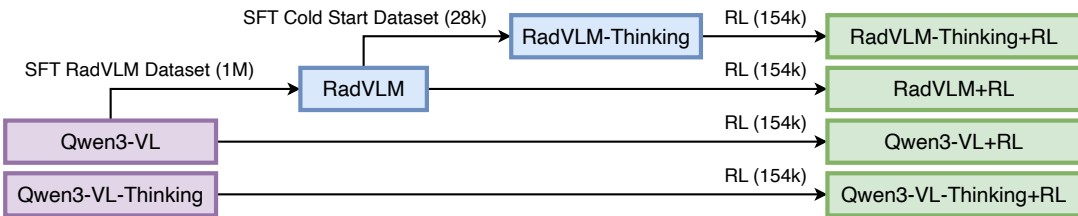

Figure 2: **Overview of Model Checkpoints and Training Pipeline.** The diagram illustrates how different model variants are initialized and refined with SFT and RL.

token `</think>` and collect a total of around 28k such data points, including 19k for report generation and 9k for visual grounding.

### 3.3. Model Training

We train multiple models in stages. We first create SFT checkpoints (RadVLM and RadVLM-Thinking) and then enhance these checkpoints with RL. Figure 2 summarizes the SFT and RL pipeline. Throughout, Qwen3-VL refers to the 8B Instruct model.

We begin by performing SFT on the RadVLM dataset (see Section 3.2). Starting from Qwen3-VL-8B-Instruct, we finetune on the question-answer pairs, yielding an updated version of RadVLM, as visualized in Figure 2 (see Appendix 5 for comparison to the original RadVLM model). Based on this SFT checkpoint, we perform a second SFT stage, where we combine datapoints from the cold-start dataset with a small subset of direct question-answer datapoints, resulting in RadVLM-Thinking. The cold-start datapoints have a `/think` suffix to differentiate them. The data mix between cold-start and SFT datapoints in that phase is designed to prevent potential catastrophic forgetting.

Following the SFT phase, we continue with RL to further optimize our model and align it with radiology-specific objectives. We perform GRPO training on continuous, task-specific rewards. For all models, the reward is computed on the final answer. For thinking models, the final answer is defined as the text following the `</think>` delimiter; if no such final answer is produced, a reward of 0 is assigned, unless noted otherwise. For non-thinking models, the entire model output is treated as the final answer.

Given our multitask setting (report generation and visual grounding), we design rewards aligned with both tasks. For report generation, we use as a reward RadCliQ (Yu et al., 2023), a composite metric consisting of BERTScore (Zhang et al., 2020), CheXbert (Smit et al., 2020) vector similarity, and RadGraph-F1 (Jain et al., 2021; Delbrouck et al., 2024), which has been shown to align better with experts than any of these alone (Yu et al., 2023).

For visual grounding, we use a *Hungarian-matched soft-F1* reward that provides smooth, per-example feedback for bounding-box alignment. For each image, we compute pairwise IoUs between predicted and reference boxes and obtain a one-to-one assignment via the Hungarian algorithm. Matched predictions contribute proportionally to their IoU (soft true positives), while unmatched predictions and ground-truth boxes act as false positives and false negatives. In contrast to mAP, which counts a prediction as correct only if its IoU exceeds a fixed threshold, this reward gives partial credit for close matches and avoids

discontinuities, offering a more stable and informative signal for RL. Our design follows DETR's matching strategy (Carion et al., 2020) and draws inspiration from IoU-weighted metrics such as Panoptic Quality (Kirillov et al., 2019). To our knowledge, this reward formulation has not been explored in prior VLM research; we introduce it here as a novel and effective optimization signal to guide future work in grounded reinforcement learning.

## 4. Experiments

### 4.1. Training settings

We start with Qwen3-VL-8B-Instruct (Bai et al., 2025a) and perform SFT on the entire RadVLM instruction dataset for two epochs with a learning rate of $1e - 5$, train batch size of $8 \times 64$ (64 GPUs), using LLaMA-Factory (Zheng et al., 2024), resulting in an updated RadVLM requiring approximately 100 GPU-hours. Next we add thinking capabilities to RadVLM by performing another round of SFT with the same settings on a mix of the cold-start dataset and RadVLM instruct dataset to prevent catastrophic forgetting.

Finally, we perform GRPO individually for each task and model variant, with 300 steps and a batch size of 512 prompts, processing a total of 154k processed prompt-image pairs. We create 8 rollouts per prompt-image pair, the KL penalty coefficient is 0.01, and we use asymmetric clipping (Yu et al., 2025). One training run takes between 350 and 1000 GPU-hours depending on model variant. Please refer to Appendix Table 7 for a detailed overview of hyperparameters. To apply GRPO (Shao et al., 2024) we use the verl library (Sheng et al., 2024). As rewards, we use RadCliQ for report generation and soft-F1 for grounding.

For inference and rollouts generation, we use vLLM (Kwon et al., 2023), a library optimized for LLM inference. For reference, RadVLM+RL processes 44.2 samples/s and RadVLM-Thinking+RL processes 11.2 samples/s (due to its additionally generated CoT), on one GH200 node with four GPUs.

We use the RadCliQ implementation provided in RadEval (Xu et al., 2025), with RadGraph-XL (Delbrouck et al., 2024) as our chosen RadGraph model. Since lower is better for RadCliQ, we use -RadCliQ as a reward. Note that $-$RadCliQ frequently computes values lower than 0 and thus we provide a reward of $-3$ if there is no final answer. Since RadGraph does not support batch inference, we spin up 32 individual copies of RadCliQ such that the reward computation does not become a bottleneck.

### 4.2. Baseline models

We evaluate a broad set of baseline models, including Qwen3-VL variants (Yang et al., 2025), CheXagent-2 (Chen et al., 2024), MedGemma (Sellergren et al., 2025), DeepMedix-R1 (Lin et al., 2025), MAIRA-2 (Bannur et al., 2024), and Llava-Rad (Zambrano Chaves et al., 2025). We denote our GRPO-trained variants with "+RL", and "Thinking" models generate intermediate reasoning before providing the final answer. The prompts and Hugging Face (HF) links are provided in Appendix G and C respectively.

### 4.3. Evaluation

We evaluate on the report generation test set of RadVLM, derived from MIMIC-CXR (Deperrois et al., 2025a). The test set consists of single frontal chest X-rays paired with findings,

with references to prior studies removed. We use various NLP and clinical metrics to evaluate report generation (see Appendix A for details). To quantify textual similarity, we report BERTScore (Zhang et al., 2020) and ROUGE-L (Lin, 2004). To evaluate clinical correctness, we report RadGraph F1 (Jain et al., 2021) using RadGraph-XL (Delbrouck et al., 2024), CheXbert F1 (Smit et al., 2020), RadCliQ (Yu et al., 2023) and GREEN (Ostmeier et al., 2024).

For the grounding task, we report mAP@0.5, computed by considering a prediction correct if its IoU with an unmatched reference box exceeds 0.5, in line with previous work on object detection benchmarks (Everingham et al., 2010; Lin et al., 2014). In particular, we report this metric on the test splits for anatomical grounding (Chest Imagenome dataset), abnormality grounding (VinDr-CXR dataset) and phrase grounding (MS-CXR and PadChest-GR datasets), following the same procedure as in Deperrois et al. (2025b). For thinking models, we extract the final answer following `</think>` and evaluate it using the same procedure as for direct answers.

Although we employ a wide range of metrics, we did not conduct human expert evaluation that could provide further insight into the clinical utility of our models.

## 4.4. Results

### 4.4.1. GRPO CONSISTENTLY IMPROVES RADVLM AND QWEN3-VL PERFORMANCE

For report generation (Table 1), GRPO substantially improves performance for both RadVLM and Qwen3-VL models. Notably, the held-out metric GREEN improves, indicating robustness with respect to the reward signal. Among all evaluated models, GRPO-optimized RadVLM variants achieve the strongest overall performance. However, despite these gains, RadVLM models remain behind in terms of CheXbert-macro. For Qwen3-VL variants, applying GRPO alone leads to large improvements, even outperforming the SFT-only RadVLM variant in some metrics. Comparing thinking and non-thinking variants, thinking models generally perform slightly worse than their instruct counterparts, with the exception of RadVLM+RL in report generation where both variants achieve comparable performance. In Figure 3a, we provide qualitative comparisons between RadVLM and RadVLM+RL. The RL-optimized model demonstrates superior clinical alignment by: correctly explicitly stating negative findings, such as the absence of pulmonary edema (top); capturing critical pathologies like effusions that the base model misses in favor of excessive device descriptions (middle); and reducing hallucinations, avoiding incorrect mentions of pleural effusions or aortic tortuosity observed in the base RadVLM predictions (bottom).

For visual grounding (Table 2), general-domain models perform poorly on CXR grounding, with a small but consistent advantage for the non-thinking variant – in line with Qwen3-VL's grounding performance reported in Bai et al. (2025a). After GRPO training, both variants improve across all tasks, again with the non-thinking model ahead, indicating that RL alone can strongly enhance out-of-domain grounding – yet they still remain slightly below SFT-only RadVLM. Applying RL on top of RadVLM further increases mAP, showing that even a well-trained in-domain model benefits from task-aligned RL. In contrast, the RL-optimized thinking variant of RadVLM does not outperform its direct-answer counterpart. In Figure 3b , we visualize predicted bounding boxes on chest X-rays, illustrating a

Table 1: **Report generation performance**. ROUGE-L (R-L), BERTScore (B-S), CheXbert micro and macro (CXb), RadGraph F1 (RGF1), GREEN (GRN), and RadCliQ (RCQ) are evaluated on the report generation test set of RadVLM, derived from MIMIC-CXR. For RadCliQ, lower is better ($\downarrow$); for all other metrics, higher is better. RL denotes training with GRPO. All models are evaluated using our pipeline. **Bold** values indicate the highest performance, while underlined values represent the second-best performance. Evaluation prompts are listed in Appendix G.

| Model | Size (B) | R-L | B-S | CXb-micro | CXb-macro | RGF1 | GRN | RCQ$\downarrow$ |
|---|---|---|---|---|---|---|---|---|
| MedGemma-pt | 4 | 20.7 | 47.7 | 49.8 | 32.5 | 15.5 | 21.9 | 1.37 |
| MedGemma-it | 27 | 15.9 | 31.3 | 47.0 | 31.5 | 12.0 | 23.3 | 1.79 |
| MAIRA-2 | 7 | 17.7 | 46.6 | 52.1 | 35.8 | 12.9 | 21.3 | 1.42 |
| CheXagent-2 | 3 | 22.5 | 37.4 | 54.5 | 38.7 | 20.1 | 29.9 | 1.45 |
| DeepMedix-R1 | 7 | 22.3 | 52.8 | 48.2 | 28.3 | 18.6 | 30.0 | 1.23 |
| LLaVA-Rad | 7 | 22.2 | 48.9 | 53.3 | **39.2** | 16.8 | 28.6 | 1.34 |
| Qwen3-VL | 8 | 14.0 | 42.0 | 35.3 | 20.4 | 10.8 | 21.9 | 1.67 |
| Qwen3-VL-Thinking | 8 | 15.0 | 43.1 | 34.6 | 18.5 | 10.9 | 18.5 | 1.63 |
| Qwen3-VL+RL | 8 | 24.7 | 55.8 | 47.4 | 22.3 | 22.5 | 25.8 | 1.05 |
| Qwen3-VL-Thinking+RL | 8 | 23.3 | 55.0 | 44.4 | 22.4 | 21.0 | 24.7 | 1.12 |
| RadVLM | 8 | 26.0 | 53.3 | 49.0 | 30.5 | 19.0 | 29.1 | 1.14 |
| RadVLM+RL | 8 | 29.9 | **59.2** | **57.0** | 33.4 | **25.8** | 32.7 | **0.86** |
| RadVLM-Thinking+RL | 8 | **30.0** | 59.0 | 56.3 | 33.6 | 25.7 | **32.9** | 0.87 |

progressive refinement in localization: the loose or misplaced regions predicted by the base Qwen3-VL are visibly corrected by SFT, and further tightened by RL.

#### 4.4.2. REWARD CHOICE INFLUENCES REPORT GENERATION QUALITY AND LENGTH

We explore the impact of the reward function on the report generation task, comparing RadCliQ, BERTScore, RadGraph-F1, and GLEU (Wu et al., 2016) in Table 3. We evaluate the obtained RL-optimized RadVLM models on all report generation metrics. We find that RadCliQ, BERTScore, and GLEU rewards improve all metrics, with RadCliQ ranking highest in 4 out of the 7 metrics. However, training with RadGraph-F1 only improves the corresponding metric itself and leads to very short reports, revealing strong signs of reward hacking. Since GREEN is prone to length reward-hacking (Hein et al., 2024), the higher GREEN scores of BERTScore and GLEU might be partially attributable to longer answers. Together, these results indicate that the choice of reward to optimize report generation via GRPO is crucial to avoid reward hacking and to improve performance in both lexical and clinical aspects.

Table 2: **Visual grounding performance measured in mAP (%)**. Mean average precision (mAP@0.5) scores for anatomical (Chest-Imagenome test dataset), abnormality (VinDr-CXR), and phrase grounding (Phrase$_{MS}$ for MS-CXR and Phrase$_{Pad}$ for PadChest-GR) tasks across various models. **Bold** values indicate the highest performance, while underlined values represent the second-best performance. For Qwen3-VL, we specify in the prompt that the model should output boxes in the expected format (see Appendix G).

| Model | Anatomy | Abnorm. | Phrase$_{MS}$ | Phrase$_{Pad}$ |
|---|---|---|---|---|
| MAIRA-2 | 19.8 | 11.3 | 80.1 | 38.8 |
| Qwen3-VL | 11.1 | 5.1 | 19.0 | 10.7 |
| Qwen3-VL-Thinking | 8.7 | 1.2 | 14.6 | 10.7 |
| Qwen3-VL+RL | 79.0 | 36.0 | 79.4 | 55.3 |
| Qwen3-VL-Thinking+RL | 68.8 | 27.6 | 76.1 | 42.0 |
| RadVLM | 82.1 | 44.2 | 84.6 | 59.0 |
| RadVLM+RL | 84.5 | **45.9** | **87.9** | **63.0** |
| RadVLM-Thinking+RL | **84.9** | 43.9 | 86.4 | 60.5 |

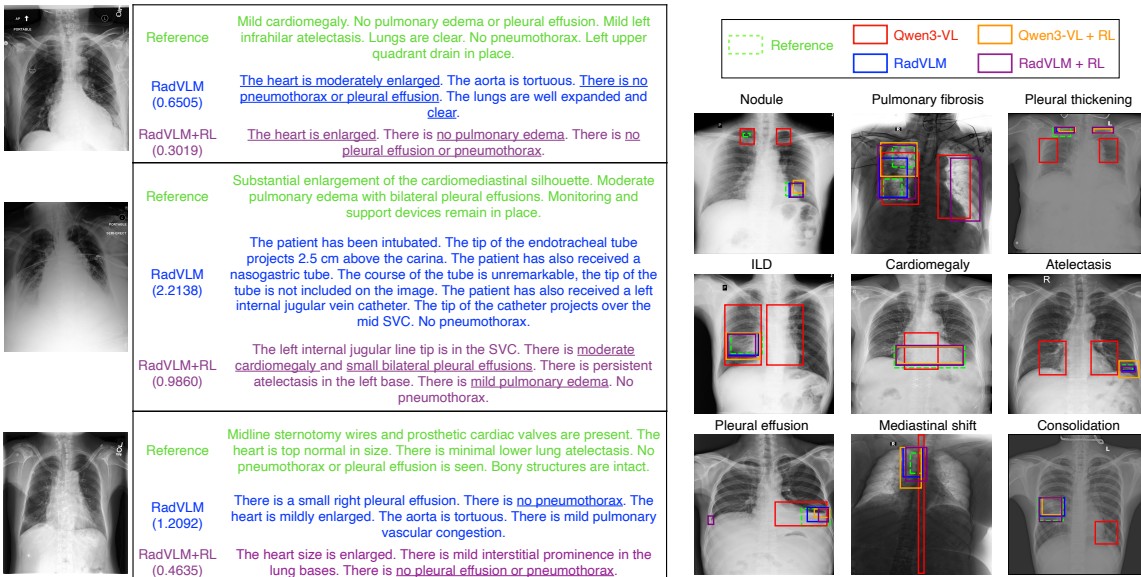

Figure 3: **Qualitative examples from baseline and RL-optimized models.** (a) Examples of generated reports from sample CXRs (top, middle, bottom) with corresponding RadCliQ score. Underlined phrases match the reference, allowing for differences in severity and excluding support devices. (b) Abnormality grounding on sample CXRs, plotting bounding boxes from different models against the reference. We provide output examples of RadVLM-Thinking+RL in Appendix K.

Table 3: **Effect of reward choice on report generation performance**. We compare RadVLM+RL with four different rewards, namely RadCliQ (RCQ), BERTScore (B-S), RadGraph-F1 (complete; RGF1-C), and GLEU across all evaluation metrics. The setting is the same as in Table 1, with the addition of the average number of characters in the response (ANC). **Bold** values indicate the highest performance, while underlined values represent the second-best performance, and ∗ indicates that the evaluated metric is used as a training reward.

| Model | R-L | B-S | CXb-micro | CXb-macro | RGF1 | GRN | RCQ↓ | ANC |
|-------|-----|-----|-----------|-----------|------|-----|------|-----|
| RadVLM | 26.0 | 53.3 | 49.0 | 30.5 | 19.0 | 29.1 | 1.14 | 207 |
| + RCQ | **29.9** | 59.2 | **57.0** | 33.4 | **25.8** | 32.7 | **0.86**∗ | 171 |
| + B-S | 29.0 | **61.6**∗ | 56.5 | **35.1** | 23.9 | 33.1 | 0.93 | 206 |
| + RGF1-C | 22.8 | 48.8 | 34.1 | 18.4 | 20.1∗ | 23.5 | 1.20 | 93 |
| + GLEU | 29.8 | 59.2 | 56.0 | **35.1** | 25.3 | **33.7** | 0.92 | 209 |

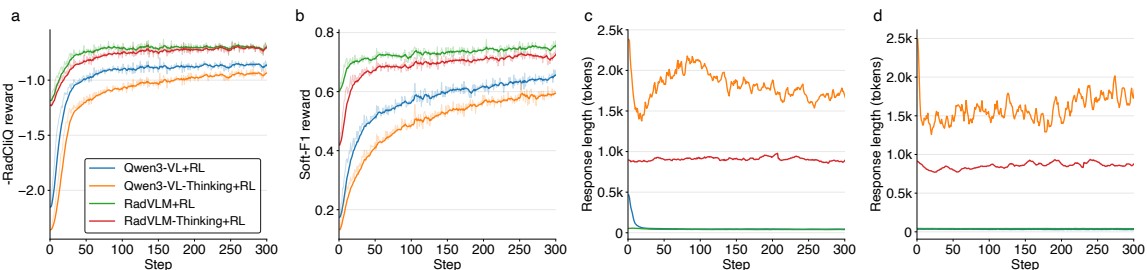

Figure 4: **Training dynamics with GRPO.** Training rewards and response lengths are shown for report generation (a, c) and visual grounding (b, d) as a function of RL steps.

### 4.4.3. THINKING MODELS SHOW SLOWER REWARD CONVERGENCE

We plot the evolution of the reward and response length during the RL training of RadVLM and Qwen3-VL models in Figure 4. We observe that the rewards of the Thinking variant (orange and red curves) remain slightly lower than the non-thinking versions, before eventually catching up. The RadVLM models (red and green) start out with a higher reward than Qwen3-VL models due to their initial in-domain SFT stage. For report generation, we notice that Qwen3-VL+RL (Figure 4c, blue curve) initially generates long reports and become shorter after a few RL steps, due to the RadCliQ optimization that discourages unwanted statements. Finally, thinking models tend to generate a relatively stable amount of tokens over RL steps (Figure 4c-d).

## 5. Discussion

In this work, we explored how RL can enhance the report generation and visual grounding capabilities of a foundation model for radiology. We show that by careful choice of rewards, the GRPO algorithm can further optimize RadVLM on tasks on which the SFT version was already performing well. This demonstrates that medical VLMs can be subjected to clinically relevant optimization enabled by modern RL techniques, overcoming limitations of SFT. Using the same optimization process, we also explored the potential benefits of thinking and found that in our settings, direct answers provide similar or better performance with lower inference cost.

A potential limitation of our study is the choice of cold-start generation strategy. We chose to generate CoT by providing the prompt combined with the image and the expected ground truth, potentially biasing the model toward generating CoT that directly leads to that specific answer. However, we also explored other strategies that do not provide the ground truth and filter generated CoT-answer pairs based on the performance of the answer (see Appendix I). We observed that the thinking model underperforms compared to the non-thinking variant, consistent with Qwen3-VL on some tasks such as 2D/3D grounding (Table 10, Bai et al., 2025a) and related work (Li et al., 2025a; Lai et al., 2025). Many factors could be at play here: initial performance degradation due to cold-start fine-tuning, non-informative thinking traces for the tasks at hand, lack of vision modality in the thinking process, reward functions not explicitly encouraging faithful reasoning steps, or lack of radiology content of VLMs such as Qwen3-VL as compared to math or code domains (Cheng et al., 2025).

In this vein, future work could involve curating examples of thinking traces designed by expert radiologists and optimizing multi-turn conversations that include questions, feedback, and tools with which the model can manipulate images (cropping, zooming, drawing) and re-observe (Zheng et al., 2025b; Su et al., 2025; Bai et al., 2025a). Regardless of the underlying cause, we believe that reporting these negative results is beneficial for the field. By demonstrating that current thinking strategies might not necessarily yield gains in radiology VLMs compared to direct-answer approaches, we encourage future studies to also disentangle the benefits of RL from the impact of reasoning traces.

Another path for further exploration is LLM-as-Judge rewards, such as GREEN (Ostmeier et al., 2024). LLM-as-Judge approaches require additional care to prevent reward hacking (Hein et al., 2024; Zhao et al., 2025), to design robust prompts, and to select appropriate models (Li et al., 2025b). Here, we rely on a simpler, yet powerful metric RadCliQ, to perform RL-optimization, as it covers lexical and clinical aspects, and is not subject to verbose hacking (as shown by report length decrease in Table 3), yet still improving metrics that favor longer reports, such as GREEN. Yet, LLM-as-Judge is a powerful tool and could potentially enable optimization of multi-turn conversations by simultaneously serving as the reward mechanism and a conversational partner for the model under optimization.

Overall, by performing large-scale training on CXR data, our study highlights the importance of cooperation between fine-tuning and reinforcement learning in the development of medical VLMs. Although some questions remain open, we hope that our findings offer a solid foundation for future work on optimizing clinically aligned multimodal models.

## Acknowledgments

This work was supported as part of the Swiss AI Initiative by a grant from the Swiss National Supercomputing Centre (CSCS) under project ID a135 on Alps. BG received support from the Swiss National Science Foundation (SNSF) grant 10003518. ND received support from RADICAL (Project-Call 2024.1, ID:9), funded by the DIZH.

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

## Appendix A. Evaluation Metrics Details

- **BERTScore** (Zhang et al., 2020): computes token-level semantic similarity using contextual embeddings from a pretrained model (see Table 8).

- **ROUGE-L** (Lin, 2004): evaluates the longest common subsequence between generated and reference texts.

- **RadGraph F1** (Jain et al., 2021; Delbrouck et al., 2024): measures structural overlap between predicted and ground-truth entities and their relations. We employ the **RadGraph-XL** (Delbrouck et al., 2024) model and report the partial reward variant of the metric.

- **CheXbert F1** We report the macro F1 over the 14 labels.

- **RadCliQ** (Yu et al., 2023) is a composite metric which first standardizes and then linearly combines RadGraph F1, CheXbert vector similarity, BERTScore, and BLEU to align better with radiologists. We use RadGraph-XL (Delbrouck et al., 2024) as our chosen RadGraph model.

- **GREEN** (Generative Radiology Report Evaluation and Error Notation) (Ostmeier et al., 2024): an LLM-as-Judge metric that identifies clinically significant errors and matched findings in generated reports.

## Appendix B. Comparison between original and our updated RadVLM.

We compare our updated RadVLM to the original RadVLM in Table 4 and Table 5. The updated RadVLM, based on Qwen3-VL, outperforms the original RadVLM in all metrics except for the anatomy visual grounding.

Table 4: **Updated RadVLM comparison Report Generation.** We compare the original RadVLM (Deperrois et al., 2025b) to the updated RadVLM based on Qwen3-VL on report generation.

| Model | Size (B) | R-L | B-S | CXb-micro | CXb-macro | RGF1 | GRN | RCQ↓ |
|---|---|---|---|---|---|---|---|---|
| RadVLM (original) | 7 | 25.4 | 51.9 | 46.4 | 28.9 | 18.2 | 27.7 | 1.16 |
| RadVLM (ours) | 8 | 26.0 | 53.3 | 49.0 | 30.5 | 19.0 | 29.1 | 1.14 |

Table 5: **Updated RadVLM comparison Visual Grounding.** We compare the original RadVLM (Deperrois et al., 2025b) to the updated RadVLM based on Qwen3-VL on visual grounding.

| Model | Anatomy | Abnorm. | Phrase$_{MS}$ |
|---|---|---|---|
| RadVLM (original) | 85.8 | 34.6 | 81.8 |
| RadVLM (ours) | 82.1 | 44.2 | 84.6 |

## Appendix C. Hugging Face Links

We link to the model weights in Table 6.

Table 6: **Model Links.** Huggingface links for the models we use.

| Model | Hugging Face |
|---|---|
| Qwen3-VL-8B-Instruct | https://huggingface.co/Qwen/Qwen3-VL-8B-Instruct |
| Qwen3-VL-8B-Thinking | https://huggingface.co/Qwen/Qwen3-VL-8B-Thinking |
| Qwen3-VL-235B-A22B-Instruct | https://huggingface.co/Qwen/Qwen3-VL-235B-A22B-Instruct-FP8 |
| CheXagent-2-3b | https://huggingface.co/StanfordAIMI/CheXagent-2-3b |
| MedGemma-4b-pt | https://huggingface.co/google/medgemma-4b-pt |
| MedGemma-27b-it | https://huggingface.co/google/medgemma-27b-it |
| DeepMedix-R1 | https://huggingface.co/Qika/DeepMedix-R1 |
| MAIRA-2 | https://huggingface.co/microsoft/maira-2 |
| Llava-Rad | https://huggingface.co/microsoft/llava-rad |

## Appendix D. verl parameters

We list the parameters we use for verl in Table 7.

Table 7: **Parameters used with verl.** If there are two values then the first one is for no thinking and the second one is for with thinking.

| Group | Name | Value |
|---|---|---|
| Data | batch_size | 512 |
| | max_prompt_length | 4096 |
| | max_response_length | 1024 / 4096 |
| | min_pixel | 1024 |
| | max_pixel | 451584 |
| Actor | lr | 1e-6 |
| | ppo_mini_batch_size | 128 |
| | ppo_micro_batch_size_per_gpu | 4 |
| | use_kl_loss | True |
| | kl_loss_coef | 0.01 |
| | kl_loss_type | low_var_kl |
| | clip_ratio_low | 0.20 |
| | clip_ratio_high | 0.28 |
| Rollout | log_prob_micro_batch_size_per_gpu | 16 |
| | n: group size per prompt | 8 |
| | log_prob_micro_batch_size_per_gpu (ref) | 16 |
| Trainer | n_gpus_per_node | 4 |
| | nnodes | 8 |
| | save_freq | 20 |
| | test_freq | 20 |

## Appendix E. BERTScore settings

We list the BERTScore setting in Table 8.

Table 8: **Settings used with BERTScore.**

| Name | Value |
|---|---|
| model_type | distilbert-base-uncased |
| num_layers | 5 |
| all_layers | False |
| idf | False |
| lang | en |
| rescale_with_baseline | True |

## Appendix F. Model inference throughput

Table 9: **Model inference throughput.**

| Model | Throughput (samples/s) | vLLM | Task |
|---|---|---|---|
| RadVLM+RL | 45.7 | Yes | Report Generation |
| RadVLM-Thinking+RL | 11.3 | Yes | Report Generation |
| CheXagent-2 | 1.08 | No | Report Generation |
| DeepMedix-R1 | 0.12 | No | Report Generation |
| DeepMedix-R1 | 20.4 | Yes | Report Generation |
| RadVLM+RL | 42.7 | Yes | Visual Grounding |
| RadVLM-Thinking+RL | 11.2 | Yes | Visual Grounding |
| RadVLM+RL | 44.2 | Yes | Both |
| RadVLM-Thinking+RL | 11.2 | Yes | Both |

## Appendix G. Prompts for Models

We list the prompts we use for the various models. Summary of the input prompts (or templates / description of how the inputs are processed) employed for inference by each compared model. These templates were obtained from released material (paper, GitHub repo, etc.) when available; otherwise, RadVLM's prompts were used (for LLaVA-OV and LLaVA-Med). "HF" denotes HuggingFace. For all RadVLM variants and for the RL training of Qwen3-VL models we use the corresponding RadVLM prompts.

### G.1. Report Generation

> **CheXagent-2**
>
> Write an example findings section for the CXR

> **MAIRA-2**
>
> HF template: processed inputs passed directly to model (no explicit prompt)

> **DeepMedix-R1**
>
> <image> Please act as an experienced radiologist and generate the "FINDINGS" section of an X-ray report based on the provided image(s). Carefully examine the image(s) and describe all observed anatomical structures and abnormalities in a systematic and objective manner. You FIRST think about the reasoning process as an internal monologue and then provide the final answer. The reasoning process MUST BE enclosed within <think> </think> tags. During this reasoning process, prioritize analyzing the local regions of the image by leveraging the bounding box coordinates in the format [x_min, y_min, x_max, y_max]. The final answer MUST BE put in \boxed{}. An example is like: <think> reasoning process 1 with [x_min1, y_min1, x_max1, y_max1]; reasoning process 2 with [x_min2, y_min2, x_max2, y_max2] </think>. The answer is: \boxed{answer}.

> **LLaVA-Rad**
>
> <image>\nDescribe the findings of the chest x-ray.\n

> **google/medgemma-27b-it**
>
> <image> You are an expert radiologist. Please succinctly describe the findings for the above chest x-ray.

**google/medgemma-4b-pt**

<image> findings:

**Qwen3-VL-8B-Instruct and Qwen3-VL-8B-Thinking**

<image>\nPlease write a radiology report for this Chest X-ray.\n\nIt should be one unstructured paragraph of findings only: concise, natural clinical language, objective, declarative sentences describing visible features only, suitable for a radiology findings report using standard radiology phrasing.

## G.2. Visual Grounding

**MAIRA-2**

HF template: processed inputs passed directly to model

**Qwen3-VL-8B-Instruct and Qwen3-VL-8B-Thinking**

<image>\nCan you tell me where [abnormality] is? Answer with bounding boxes only in the format $[x_{min}, y_{min}, x_{max}, y_{max}]$, values normalized between 0 and 1 and rounded to two decimals. Multiple boxes should be separated by 'and'.

## Appendix H. Qwen3-VL-235B Performance on Tasks

We create an overview of the performance of Qwen3-VL-235B models listed in the Qwen3-VL technical report (Bai et al., 2025a) in Table 10.

Table 10: Summary of Qwen3-VL-235B-Thinking and -Instruct results across benchmark groups. Scores are taken from the Qwen3-VL technical report (Bai et al., 2025a) and directly averaged within each group to provide a high level overview, despite differences in scales across benchmarks. Further, number of individual benchmarks that favor Instruct, Thinking, or Tie respectively are listed.

| Benchmark | Instruct | Thinking | Favor In-struct | Favor Think-ing | Ties |
|---|---|---|---|---|---|
| STEM Puzzle | 62.31 | **66.49** | 1 | **13** | 0 |
| General VQA | **79.76** | 79.74 | **3** | 2 | 0 |
| Alignment | 54.33 | **55.97** | 0 | **2** | 1 |
| Document Understanding | **131.25** | 128.11 | **6** | 5 | 2 |
| 2D/3D Grounding | **57.13** | 54.77 | **4** | 2 | 0 |
| Embodied/Spatial Understanding | 66.40 | **68.12** | 1 | **4** | 0 |
| Multi-Image | 71.85 | **73.60** | 1 | 1 | 0 |
| Video Understanding | 73.61 | **73.74** | 5 | 2 | 0 |
| Perception with Tool | **87.17** | 82.27 | **3** | 0 | 0 |
| Multi-Modal Coding | **80.77** | 79.20 | **2** | 1 | 0 |
| Multi-Modal Agent | 50.58 | **52.46** | 2 | **3** | 0 |
| Average | 79.26 | **79.43** | 28 | **35** | 3 |

## Appendix I. Cold-Start Prompt

### I.1. Report Generation

We provide the prompt to generate the cold-start dataset. The "guide" (GUIDE) is adapted based on sections 1.2 - 1.10 of https://www.ncbi.nlm.nih.gov/books/NBK553874/ with tables converted into markdown. The guide and ground truth are substituted into the prompt in their respective placeholders. Note that some whitespace has been removed in this box.

---

**Report Generation Cold-Start Prompt**

You are an expert thoracic radiologist analyzing a chest radiograph.
Your task is to perform a **structured, detailed, and systematic reasoning process** based on the GUIDE below and the GROUND TRUTH report.
—

### INSTRUCTIONS
1. **Follow the GUIDE carefully and comprehensively.** - Proceed systematically through every anatomic and technical category described in the GUIDE. - For each category, analyze findings in a detailed, clinically reasoned manner.
2. **Use of the Ground Truth** - Treat each *positive* finding explicitly stated in the Ground Truth as present with the same certainty level. - Treat each *negative* finding explicitly stated in the Ground Truth as absent with the same certainty level. - Treat each *uncertain* finding in the Ground Truth as indeterminate with the same certainty level. - You must explicitly mention each item in the Ground Truth in the appropriate section. Do not skip any item, even if it seems incidental. - **Crucially: if a finding is *not mentioned* in the Ground Truth, do *not* assume absence.** Silence in the Ground Truth means the status is unknown until you assess the image. - Base your reasoning primarily on the image. If the image evidence is insufficient, label the finding as **indeterminate** rather than inferring from Ground Truth silence. - The output must not mention or allude to having access to the Ground Truth.
3. **Certainty Language** - When appropriate, qualify statements as one of: - **Present (positive)** - clear imaging evidence supports the finding. - **Absent (negative)** - clear imaging evidence supports absence. - **Indeterminate/uncertain** - imaging is equivocal, limited, or non-diagnostic for that point. - **Not assessable** - technical factors preclude evaluation. - Choose the least speculative label when evidence is limited.
4. **Output Requirements** - Provide **only your structured reasoning process**, not a final radiology report. - The reasoning should be clear, logically structured, and clinically grounded. - Do not mention instructions, the GUIDE, or any external references. - Do not reference the ground truth in the structured reasoning process. Write as if directly interpreting the image. - Discuss the following in order, using paragraph form (not lists): - **Technical quality** (positioning, penetration, motion, lung volumes, artifacts) - **Support and monitoring devices and other foreign bodies/surgical materials** - **Chest wall** - **Mediastinum** (heart, great ves-

sels, masses, lines/stripes/interfaces, calcification, pneumomediastinum) - **Hila**
- **Lungs** (volumes, atelectasis, air space opacities, interstitial changes, nodules/-masses, abnormal lucency) - **Airways** - **Pleura and diaphragm**
5. **Tone and Style** - Be explicit, methodical, and clinically precise. - Use full sentences and natural medical phrasing. - Demonstrate radiologic reasoning rather than just listing facts.
—
### GUIDE
{GUIDE}
—
### GROUND TRUTH
{GROUND_TRUTH}

## I.2. Visual Grounding

We provide the prompt to generate the cold-start dataset for visual grounding. The relevant parts are substituted into the prompt in their respective placeholders. Note that some whitespace has been removed in this box.

---

**Report Generation Cold-Start Prompt**

### System You are an expert chest-radiology vision-language model specialized in **visual grounding**. Your main goal is to produce an extensive, detailed reasoning trace that describes **how you locate the visual evidence** in the chest X-ray that supports the question, before revealing the answer.
For every case you will receive - one chest X-ray image - a clinical question about that image - the ground-truth answer to that question
**Your task**
1. Carefully examine the image and reason **step-by-step** about the **spatial evidence** related to the question. - Focus on **where** the relevant anatomical structure or abnormality is located. - Describe **which region(s)** of the image you examine, how you navigate the X-ray, and how you narrow down to specific areas. - Explain spatial relationships (left/right, upper/mid/lower zones, central/peripheral, anterior/posterior) and link findings to nearby anatomical landmarks (diaphragm, heart border, ribs, hilum, costophrenic angles, etc.). - Discuss the **bounding-box level reasoning**: which portion of the image would contain the key feature. - Mention what visual patterns guide your search (opacity, lucency, silhouette loss, contour shape, asymmetry). - Consider alternative locations or causes and justify why you include or exclude them. - Be **very detailed and verbose** - the reasoning should read like a radiologist's internal monologue mapping the image to bounding boxes. - You must complete this reasoning **fully** before revealing the answer.
2. Write this full reasoning inside one continuous block that ends with **`</think>`**. - Do **not** output the answer or hint at it before this token. - The reasoning block should be long, coherent, and spatially rich.

3. Immediately after '</think>', output the **gold answer exactly as given** in the input (copy it verbatim). - Do not add any commentary or text outside these two blocks.

—

### User
Question: {QUESTION}
Gold Answer (copy verbatim after the thinking block): {ANSWER_GT}

—

### Assistant <!– your richly detailed spatial reasoning trace describing how you identify the relevant region(s) –> </think> {ANSWER_GT}

# Appendix J. Distilling Cold-Start from Qwen3-VL-32B-Thinking

We also tried distilling from Qwen3-VL-32B-Thinking. While these results are preliminary, we think they might be helpful for future work.

## J.1. Report Generation

In addition to the cold-start generation method where we provide the ground truth, we also tried to distill from Qwen3-VL-32B-Thinking by sampling 8 times per image (without providing the ground truth) and then selecting the best answer, together with its corresponding thinking trace, according to RadCliQ. First, we tried a "One-Stage" approach, where we directly asked the model to output the findings in a format similar to that of our dataset. This appeared to bias the model, so we also tried a "Two-Stage" approach. We visualize the rewards in Figure 5 and observe that the approach provided with the guide and ground truth ("RadVLM-Thinking") achieves higher rewards than the "Two-Stage" approach, which in turn outperforms the "One-Stage" approach. Please keep in mind that a small percentage of the Thinking outputs are assigned a score of $-3$ because they do not produce a final answer. The evolution of the response length is shown in Figure 6. We describe the "One-Stage" and "Two-Stage" approaches next.

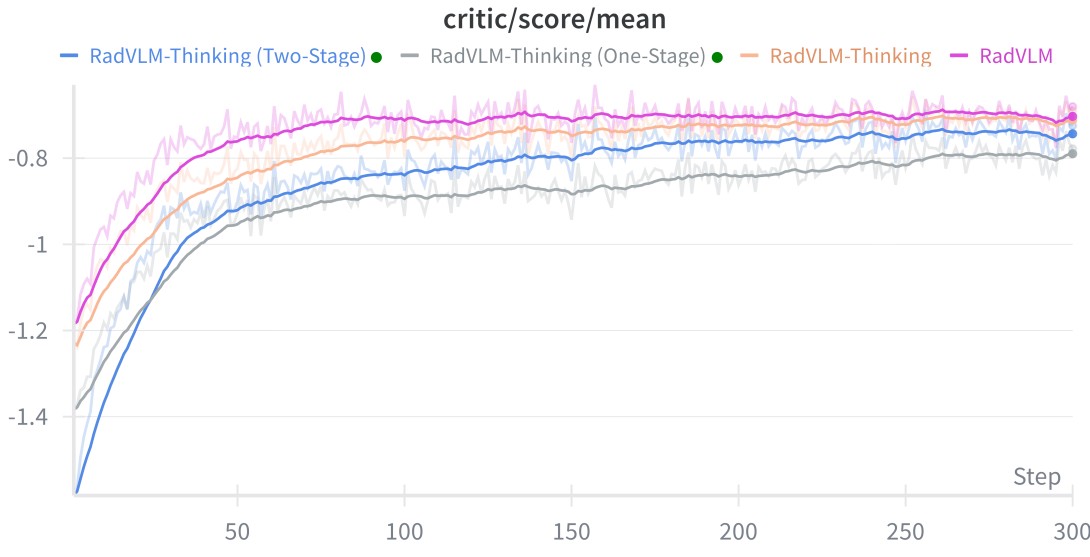

Figure 5: **Comparison of Cold-Starts: RadCliQ reward.**

### J.1.1. ONE-STAGE

For image $i$, we generate 8 responses from Qwen3-VL-32B-Thinking with the RadVLM prompts, resulting in thinking$_{i,j}$ and an answer$_{i,j}$. For image $i$ we then select the best answer$_{i,j}$ over j, according to RadCliQ, resulting in the best answer$_i$, and the corresponding thinking$_i$. We then use (thinking$_i$, answer$_i$) as a "One-Stage" cold-start datapoint.

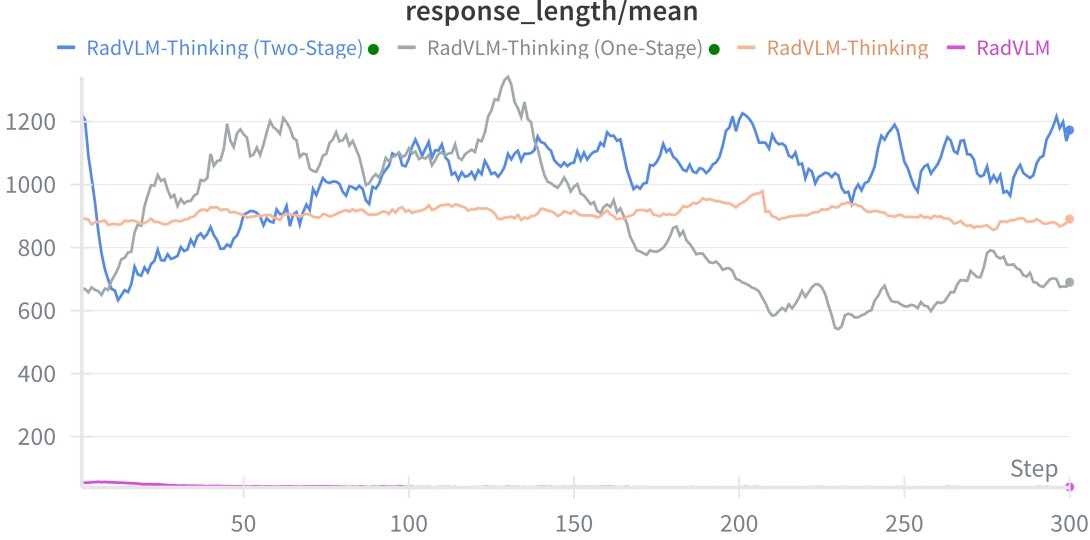

Figure 6: **Comparison of Cold-Starts: evolution of response Length.**

---

**One-Stage Prompt**

You are an expert thoracic radiologist analyzing a chest radiograph.

After finishing your reasoning, close it with </think> and then produce the final answer:
- One unstructured paragraph of Findings only: concise, natural clinical language, objective, declarative sentences describing visible features only, suitable for a radiology findings report using standard radiology phrasing.
- Include positives and relevant negatives that are visually supported.
- Use concise, compact, natural phrasing consistent with real radiology reports, keep Findings descriptive (no differential diagnoses, management, no overall judgment).
- Do not include headers, bullets, impressions, recommendations, or any mention of your thinking process.

---

### J.1.2. Two-Stage

For image $i$, we generate 8 responses from Qwen3-VL-32B-Thinking with the RadVLM prompts, resulting in thinking$_{i,j}$ and an answer$_{i,j,1}$. The answer might look totally different than what we would expect, so for every response we let Qwen3-VL-32B-Instruct rewrite answer$_{i,j,1}$ into answer$_{i,j,2}$ with the following Two-Stage Rewriting Prompt. For image $i$ we then select the best answer$_{i,j,2}$ over j, according to RadCliQ, resulting in the best answer$_i$, and the corresponding thinking$_i$. We then use (thinking$_i$, answer$_i$) as a "Two-Stage" cold-start datapoint.

**Two-Stage Rewriting Prompt**

You are a radiology expert.

You are given a full radiology report. Your task is to generate the Findings only.

Requirements for the Findings:
- Output exactly one unstructured paragraph of Findings text only.
- Do not include any label such as "Findings:", no headers, no section titles, no bullets, no tables.
- Do not include explanations.
- Use concise, natural clinical language with objective, declarative sentences.
- Describe only visible imaging features. Do not include differential diagnoses, do not include disease labels that are not explicitly present in the original report, do not include management, do not include overall impressions.
- Lead with devices and any urgent or acute abnormalities when present.
- Include salient positive findings and relevant negatives that are documented in the original report.
- Use standard radiology phrasing consistent with real radiology reports.
- Write the Findings in the same language as the input report.
- Do not add information that is not supported by the original report. Do not hallucinate findings.

Radiology report:
"""
{answer$_{i,j,1}$}
"""

## Appendix K. Qualitative examples of RadVLM-Thinking+RL

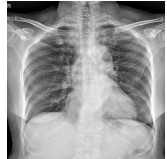

| |
|---|
| Could you specify where to find the cardiomegaly on this X-ray? |
| Concerning the cardiomegaly, you will find it at [0.35, 0.61, 0.84, 0.77] in the image. |
| To identify the location of cardiomegaly on this chest X-ray, I begin by orienting myself to the standard anatomical landmarks. The image is a posteroanterior (PA) view, as evidenced by the symmetric clavicles and the clear visualization of the thoracic spine through the cardiac silhouette — a hallmark of a well-penetrated PA radiograph. The "R" marker in the upper left corner confirms the patient's right side is on the left of the image, which is critical for accurate spatial localization.

I first scan the mediastinum, focusing on the cardiac silhouette, which occupies the central portion of the thorax, bordered laterally by the lungs and inferiorly by the diaphragm. The normal cardiac silhouette should occupy less than half the transverse diameter of the thorax. In this image, the cardiac silhouette appears enlarged, extending laterally beyond the expected boundaries — particularly on the left side, where the left heart border is displaced laterally and the cardiac apex is more prominent and rounded than usual.

To precisely localize the cardiomegaly, I mentally divide the image into quadrants and focus on the central mediastinal region. I trace the cardiac borders: the right heart border is formed by the right atrium and is visible along the right mediastinal contour, while the left heart border is formed by the left ventricle and is more prominent here. The cardiac silhouette extends from approximately the level of the 4th thoracic vertebra (T4) superiorly to the diaphragm inferiorly, and spans from the midline to the left lateral chest wall.

I then examine the cardiothoracic ratio (CTR) by comparing the transverse diameter of the heart to the internal transverse diameter of the thorax at the level of the diaphragm. Visually, the heart appears to occupy more than 50% of the thoracic width — a classic sign of cardiomegaly. The enlargement is most evident in the lower half of the cardiac silhouette, where the left ventricular contour is bulging laterally, and the cardiac apex is displaced downward and laterally, which is consistent with left ventricular enlargement.

I also assess the surrounding structures for secondary signs: the pulmonary vasculature appears mildly engorged, and there is no obvious pulmonary edema or pleural effusion, which helps confirm that the enlargement is cardiac in origin rather than due to fluid overload or other causes. The diaphragm is well-defined, and the costophrenic angles are sharp, ruling out significant pleural effusion that might obscure the cardiac borders.

The bounding box [0.35, 0.58, 0.83, 0.77] corresponds to the region of the cardiac silhouette that is most enlarged — specifically, the lower two-thirds of the heart, extending from the midline to the left lateral chest wall, and from just below the level of the aortic arch (around the 4th thoracic vertebra) down to the diaphragm. This region encompasses the left ventricle and the cardiac apex, which are the primary contributors to the increased cardiac silhouette in this case. The coordinates are normalized to the image dimensions, with 0.35 being the left boundary (approximately 35% from the left edge of the image), 0.58 the top boundary (58% from the top), 0.83 the right boundary (83% from the left), and 0.77 the bottom boundary (77% from the top). This box precisely encloses the area of maximal cardiac enlargement, which is consistent with left ventricular hypertrophy or dilation.

I also consider alternative causes of mediastinal widening, such as mediastinal mass or lymphadenopathy, but the contour is smooth and rounded, consistent with cardiac enlargement rather than a mass. There is no evidence of tracheal deviation or esophageal displacement, which further supports that the enlargement is cardiac rather than extrinsic.

In summary, the cardiomegaly is localized to the central mediastinum, specifically the lower half of the cardiac silhouette, extending laterally to the left and inferiorly to the diaphragm. The bounding box [0.35, 0.58, 0.83, 0.77] accurately captures this region of maximal cardiac enlargement, which is consistent with the visual evidence of cardiomegaly on this X-ray.
**</think>**
**The cardiomegaly is located at the coordinates [0.35, 0.58, 0.83, 0.77] on the image.** |

Figure 7: **Example of RadVLM-Thinking+RL output on abnormality grounding task.** We show the full output of our RL-optimized thinking model, including the thinking process and the following final answer (after the closing think token).

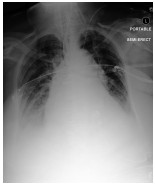

Substantial enlargement of the cardiomediastinal silhouette. Moderate pulmonary edema with bilateral pleural effusions. Monitoring and support devices remain in place.

The technical quality of this portable, semi-erect chest radiograph is limited by patient positioning and potential motion. The semi-erect posture, as indicated by the label, may result in suboptimal lung expansion and altered fluid distribution compared to a standard upright or supine view. The image demonstrates moderate to severe motion artifact, as evidenced by the blurring of the diaphragmatic contours and vascular markings, particularly in the lower lung zones. Penetration appears adequate, with faint visualization of the thoracic vertebral bodies through the mediastinum, though the overlying soft tissues and potential motion reduce contrast resolution. Lung volumes are reduced, consistent with the semi-erect position and possible underlying restrictive or dependent processes. No grid lines or detector artifacts are apparent, though the overall image quality is degraded by motion and positioning.

Multiple support and monitoring devices are present. An endotracheal tube is visualized, with its tip positioned approximately 4 cm above the carina, which is within the acceptable range for proper placement. A nasogastric tube is also present, coursing down the midline esophagus and terminating below the diaphragm, consistent with appropriate gastric placement. A central venous catheter is seen entering via the left internal jugular vein, with its tip terminating in the lower superior vena cava, near the cavoatrial junction, which is an appropriate position. No evidence of catheter malposition, kinking, or fracture is apparent. No pacemaker or defibrillator is visible.

The chest wall appears symmetric without evidence of contour deformity, swelling, or mass. No osseous abnormalities, rib destruction, or soft tissue calcification are identified. The presence of multiple monitoring leads and ECG electrodes is noted over the anterior chest wall, which is expected in a critically ill patient. No subcutaneous emphysema or air within the chest wall is seen.

The mediastinum is widened, with a prominent cardiac silhouette that appears enlarged, consistent with cardiomegaly. The cardiac borders are obscured in places, particularly on the left, suggesting pericardial or pleural effusion. The aortic arch and pulmonary artery are not clearly delineated due to overlying opacities and motion artifact. No mediastinal mass or abnormal contour is definitively identified, though the mediastinal widening may be secondary to cardiac enlargement or pericardial effusion. No mediastinal calcification or pneumomediastinum is evident. The trachea is midline without deviation or narrowing.

The hila are obscured by overlying opacities and motion artifact, making precise assessment of size, contour, or density difficult. No clear hilar enlargement or abnormal convexity is discernible. The relationship between the right and left hila cannot be reliably assessed due to the image quality and overlying opacities.

The lungs demonstrate bilateral, predominantly lower zone opacities with a ground-glass and reticular pattern, consistent with interstitial and alveolar filling processes. The opacities are more pronounced in the dependent regions, suggesting a combination of pulmonary edema and pleural effusion. No focal consolidation, mass, or nodule is clearly identified, though the image quality limits detection of small or subtle lesions. No evidence of hyperlucency or bullae is seen. The lung volumes are reduced, consistent with the semi-erect position and possible restrictive or dependent processes.

The airways, including the trachea and mainstem bronchi, appear patent without evidence of focal narrowing, stenosis, or mass. The trachea is midline without deviation. No endotracheal or endobronchial lesions are visible, though the image quality limits detailed assessment of the bronchial tree.

The pleura and diaphragm demonstrate bilateral, moderate to large pleural effusions, with blunting of the costophrenic angles and obscuration of the diaphragmatic contours, particularly on the right. The effusions appear to be free-flowing, with a dependent distribution. No pneumothorax is identified. The diaphragmatic domes are elevated, consistent with reduced lung volumes and possible diaphragmatic dysfunction or effusion-related elevation. No pleural thickening or calcification is evident. The presence of bilateral pleural effusions and pulmonary edema suggests a systemic process, such as heart failure or acute respiratory distress syndrome.
**</think>**
**The endotracheal tube tip is 4 cm above the carina. The left internal jugular catheter tip is in the stomach. Moderate bilateral pleural effusions and moderate cardiomegaly. There is mild pulmonary edema. There is no pneumothorax.**

Figure 8: **Example of RadVLM-Thinking+RL output on report generation task.** We show the full output of our RL-optimized thinking model, including the thinking process and the following final answer (after the closing think token).

