# OpenReview forum: "RadVLM-GRPO: Enhancing Chest X-ray Report Generation and Visual Grounding via Reinforcement Learning"
_MIDL.io/2026/Conference — MIDL 2026 Poster_

### Official Review · Reviewer_PHBw · 2025-12-19

**Confidence:** 3
**Preliminary Rating:** 4
**Final Rating:** 4

**Summary:**

The paper proposes a method to generate chest X-ray radiology reports using reinforcement learning with Group Relative Policy Optimization and with visual grounding based on soft-F1. The work presents a detailed literature review, experiments, comparisons with other methods in the literature, and a comparison with supervised fine-tuning.

**Strengths:**

The paper is well-written and provides extensive details and justifications of their design choices. The authors made their code publicly available. The proposed method achieved significant improvements over the methods it was compared against.

**Weaknesses:**

Although the metrics are strong and clinically motivated, no human radiologist evaluation is provided. This limits claims about real-world clinical usefulness, especially given known failure modes of automatic report metrics.

The reinforcement learning method itself (GRPO) is not novel. The contribution is primarily at the application-level.

The method is computationally heavy, trained on 64 GPUs, and doesn't present details on computational times for training and inference.

**Detailed Comments:**

- Though the title is very general, the work is only applied to chest X-rays.
- How sensitive are results to the different parameters (e.g., number of rollouts, KL coefficient, etc.)?
- Can you provide evidence that RL gains are not primarily driven by length or formatting changes (especially since some metrics are length-sensitive)?

**Justification Of Final Rating:**

I appreciate the authors for addressing most of my comments. I also agree with the main concerns from the other reviewers. My recommendation for a weak accept is mostly based on the limited novelty of the proposed model.

**Justification Of The Preliminary Rating:**

A solid paper with many experiments and comparisons. The proposed model compares favourably against other methods in the literature and supervised fine-tuning. However, the proposed model is computationally intensive and has little technical novelty.

**Questions To Address In The Rebuttal:**

Please answer the questions from the detailed comments section above.

---

> ### Author Response · Authors · 2026-01-24
>
> We thank the reviewer for the constructive feedback. In the following, we provide a detailed response and reference line numbers from the revised manuscript (new content highlighted in light blue).
>
> **Title and Scope**
>
> We agree with the reviewer that the work is specific to Chest X-rays; the title has been changed accordingly in the revised manuscript.
>
> **Hyperparameter Sensitivity**
>
> We observed on the validation set that turning off the KL coefficient results in lower performance. We also experimented with a varying number of rollouts. The training reward is slightly higher with 16 rollouts, but that implies fewer steps given a fixed budget. On the validation set, it appeared better to perform more steps to cover more images.
>
> The training cost of 350-1000 GPU-hours (mentioned l. 225-226) per RL run is a limiting factor..
>
> **Metric Sensitivity and Length Bias**
>
> We believe we can provide some evidence for this. We acknowledge human evaluation could provide deeper insights.
>
> Regarding length bias, while GREEN is length-sensitive, RadVLM+RL optimized with RadCliQ outperformed RadVLM on this metric despite having a shorter response length (Table 3). RL optimization with BERTScore and GLEU resulted in models that produced similar length outputs as RadVLM and produced higher scores than their  RadCliQ equivalent (discussed l. 352-355).
>
> Furthermore, in the three qualitative examples we added (new Figure 3), the models appear to behave differently. For example, RadVLM is verbose about support devices, while RadVLM+RL is more concise about findings (Figure 3a middle). Specifically, the sentence “the aorta is tortuous.” appears less often in RadVLM+RL outputs (42 vs 93 times, examples in Figure 3a top and bottom), suggesting the RL gains are not primarily driven by simple formatting changes but by content adjustments.
>
> **Human Evaluation and Clinical Utility**
>
> We agree that human evaluation would further express the clinical usefulness of the evaluated models. We tried to provide the most exhaustive evaluation pipeline to reflect different aspects that automatic metrics could reveal. We added some qualitative examples in the new Figure 3 and acknowledged the need for expert evaluation in the methods section (l. 263-264). We also point out the potential benefit of curating cold-start reasoning traces from experts at lines 341-344.
>
> **Technical Novelty and GRPO Application**
>
> While we agree that the contribution is application-focused, we believe it is valuable. We note the customized design of the GRPO training derived from a series of experiments leading to a specific choice of hyperparameters (rollouts, batch size, clip values, and KL divergence), all improving performance at convergence and training stability (discussed l. 90-91). We also justify the choice of GRPO among other RL optimization strategies at lines 154-158.
>
> **Computational Cost for training and inference**
>
> We have added details on computational cost. RL training runs take between 350 - 1000 GPU-hours on GH200 GPUs, depending on the settings, with thinking models taking longer due to producing more tokens (illustrated by the new Appendix K figures). SFT runs take around 100 GPU-hours (mentioned l. 219). We also added throughput information on one GH200 node with four GPUs (using vLLM, a library optimized for LLM inference): RadVLM+RL achieves 44.2 samples/second, while RadVLM-Thinking+RL achieves 11.2 samples/second (mentioned l. 229-232).

---

> > ### Comment · Reviewer_PHBw · 2026-01-27
> >
> > Thank you for your response and for addressing my comments, either by altering the paper or by acknowledging the work's limitations.

---

> > > ### Author Response · Authors · 2026-02-02
> > >
> > > We thank the reviewer for the feedback and are pleased that the revisions addressed their comments.

---

### Official Review · Reviewer_iD9C · 2025-12-31

**Confidence:** 3
**Preliminary Rating:** 3
**Final Rating:** 4

**Summary:**

This paper investigates the role of reinforcement learning (RL) and explicit intermediate reasoning (“thinking”) in vision–language models (VLMs) for Chest X-ray (CXR) interpretation. Starting from large-scale supervised fine-tuning (SFT) on CXR data, the authors build an updated RadVLM based on Qwen3-VL, introduce a cold-start SFT stage to enable basic thinking ability, and further apply Group Relative Policy Optimization (GRPO) with clinically grounded, task-specific reward functions for report generation and visual grounding. Matched experiments are conducted across domain-specific and general-domain Qwen3-VL variants, with and without thinking. The results show that while strong SFT remains essential, RL provides consistent additional gains, whereas explicit thinking does not further improve performance. Under a unified evaluation pipeline, RL-optimized RadVLM achieves state-of-the-art results on both report generation and grounding tasks. Overall, the paper provides a careful empirical study of RL and reasoning in medical VLMs, though the novelty of the RL formulation and reward design is limited, and several results require clearer contextualization and analysis.

**Strengths:**

The paper conducts a systematic and well-controlled empirical study of SFT, RL, and explicit thinking in medical VLMs.
Applying GRPO with clinically grounded reward functions to both report generation and visual grounding is practically relevant.

**Weaknesses:**

The reward functions and GRPO formulation are relatively simple and closely follow existing designs, limiting methodological novelty.
It is unclear whether performance gains arise from RL optimization itself or from better-aligned evaluation metrics.
The baseline performance of Qwen3-VL variants appears weaker than some existing medical VLMs, raising concerns about the competitiveness of the starting point.
The paper provides limited analysis on why explicit thinking degrades or fails to improve performance.

**Detailed Comments:**

Novelty of Reward Design and GRPO Application
The paper uses RadCliQ for report generation and IoU-based soft-F1 for visual grounding as reward signals in GRPO. While these rewards are clinically meaningful, both the reward formulation and the GRPO framework are relatively straightforward extensions of existing approaches. The authors should better clarify:
What aspects of the reward design are novel or specifically tailored to medical VLMs.
Why GRPO is preferred over simpler RL strategies in this setting.
Whether alternative or richer reward formulations were explored.
Strength of the Qwen3-VL Baselines
It is unclear whether the reported Qwen3-VL and Qwen3-VL-Thinking results correspond to fully trained and well-tuned baselines. Notably, their performance appears lower than some existing models such as LLaVA-Rad and CheXagent-2. The authors should clarify:
Whether all Qwen3-VL variants were trained to convergence under comparable settings.
How differences in backbone capacity, pretraining data, or fine-tuning strategies affect the comparison.
Whether the observed gains persist when starting from stronger baselines.
Effect of Explicit Thinking on Performance
A key conclusion of the paper is that explicit thinking does not improve, and may even degrade, performance in CXR tasks. However, this result deserves deeper analysis. Possible factors include:
Misalignment between thinking-style supervision and medical report generation.
Increased verbosity or deviation from clinically concise language.
Reward functions not explicitly encouraging faithful reasoning steps.
A more detailed qualitative or quantitative analysis would strengthen this claim.

**Justification Of Final Rating:**

The paper is well motivated and technically solid, with clear contributions to radiology report generation and visual grounding. The authors’ response adequately addressed my concerns. Although additional comparisons with recent VLM-based methods could further strengthen the work, the overall quality and contribution justify a positive final rating.

**Justification Of The Preliminary Rating:**

This paper presents a solid empirical study on the role of reinforcement learning and explicit reasoning in medical vision–language models, with clinically meaningful evaluation and careful experimental controls. The results are informative and relevant to the MIDL community, particularly the finding that explicit thinking does not trivially transfer to medical domains. However, the methodological novelty of the RL formulation and reward design is limited, and the baseline competitiveness and deeper analysis of key phenomena require further clarification. With improved justification, stronger baselines, and deeper insights into the failure modes of thinking, this work could become a valuable contribution.

**Questions To Address In The Rebuttal:**

Why does Qwen3-VL-Thinking underperform its non-thinking counterpart in several settings?
Can the authors provide more details on the construction and distribution of the SFT Cold Start dataset (28k) and the SFT RadVLM dataset (1M)?
How would the proposed RL framework perform with stronger or alternative reward functions?

---

> ### Author Response · Authors · 2026-01-24
>
> We thank the reviewer for the constructive feedback. In the following, we provide a detailed response and reference line numbers from the revised manuscript (new content highlighted in cyan).
>
> **Qwen3-VL-Thinking vs non-thinking counterpart**
>
> Qwen3-VL-Thinking underperforms its non-thinking counterpart in some non-CXR tasks as reported by the Qwen team (Bai et al., 2025a, Table 10), particularly outside of STEM puzzles. When evaluating Qwen3-VL variants on our radiology tasks, we observed a similar trend. Other papers (Li et al., 2025a; Lai et al., 2025) have observed that thinking models underperform their direct-answer model (referenced l. 330 - 332).
>
> This is similarly observed with RadVLM+RL and RadVLM-Thinking+RL (discussed l. 330-338, 86-87). We hope this encourages future CXR VLM research to conduct similar experiments against direct-answer + RL to disentangle gains of RL from gains of explicit thinking (discussed l. 342-346).
>
> **Cold Start dataset (28k), SFT RadVLM dataset (1M)**
>
> We have provided more details on the construction and distribution of the SFT cold start dataset (28k) and SFT RadVLM dataset (1M) in lines 160-178. A detailed overview of the SFT RadVLM dataset is provided in the original RadVLM paper (Deperrois et al., 2025). Over the 1.1M examples, 37% are for report generation, 37% for abnormality classification, 15% are for visual grounding, and 9% are for multi-turn conversations. The cold-start dataset is obtained after randomly collecting datapoints from report generation (19k) and visual grounding samples (9k).
>
> **Stronger and Alternative reward functions**
>
> In contrast to our design that solely focuses on the final answer, potentially stronger reward functions could evaluate the quality of the thinking traces themselves. Further, separate parts of rollouts could be assigned individual advantages. We conducted brief experiments with LLM-as-judge rewards which also judge the thinking trace, but found them prone to reward hacking. Future work could exploit our preliminary insights and further explore this direction. This is discussed at lines 347-355.
>
> **Novelty of Reward Design and GRPO Application**
>
> For report generation, while RadCliQ has been developed in previous work, our design required finding the most suited metric that would 1\) cover different aspects of report quality and 2\) be the least subject to reward hacking. After experiments exploring other functions (Table 3), we chose RadCliQ, noting that LLM-as-judge metrics like GREEN are prone to (length) reward hacking (discussed l. 350-353). Regarding visual grounding, the soft-F1 reward is, to our knowledge, not explored in previous work (which mostly uses mAP or IoU for 1 box); we highlight this at lines 211-213.
>
> **GRPO and other RL strategies**
>
> Regarding the choice of RL strategy, we now explain our preference for GRPO over other methods in the revised manuscript (l. 154-158). We prioritize online RL over offline methods like DPO. GRPO is a simpler and more computationally efficient online RL algorithm than PPO.
>
> **Strength of Qwen3-VL Baselines**
>
> For Qwen3-VL (without RadVLM SFT), it is expected that it underperforms on radiology tasks due to its general-domain training. However, once we perform SFT, the resulting RadVLM model performs well compared to other baselines (Table 1, Table 2). This SFT RadVLM checkpoint is our major starting point, as the paper focuses on the impact of RL on an already fine-tuned CXR VLM. Using an SFT version of Qwen3-VL provides us with an up to date model architecture and broad library support
>
> **Qwen3-VL+RL and Qwen3-VL-Thinking+RL Convergence**
>
> Regarding convergence, we fixed hyperparameters and the number of steps to 300 (l. 222-228). We plot the training rewards against steps in Figure 4.
>
> For report generation, rewards appear to plateau before step 300 for both variants. For visual grounding, while Qwen3-VL variants could potentially increase training rewards with more steps, the trend suggests Qwen3-VL-Thinking+RL would remain lower than Qwen3-VL+RL, which is why we decided not to spend more GPU-hours on these runs.
>
> **Stronger baselines than RadVLM**
>
> While we did not explore stronger starting baselines than RadVLM, results show that although dramatically improving, Qwen3-VL+RL still converges to a lower performance than RadVLM+RL. This indicates that a stronger baseline likely results in higher performance, even though RL tends to close the gap (Figure 4).
>
> **Thinking analysis**
>
> We agree that this deserves deeper analysis. We thank the reviewer for pointing out the possible factors for that thinking degrades performance.
>
> In this paper we would like to report the observation that certain cold start strategies on a CXR VLM result in models that after RL (RadVLM-Thinking+RL) strongly outperform the initial model (RadVLM) yet underperform or only match the performance of the equivalent direct answer model trained with RL (RadVLM+RL) (discussed l. 342-346, l. 86-87).

---

> > ### Comment · Reviewer_iD9C · 2026-02-02
> >
> > Thanks for the authors’ detailed response. I think my concerns have been addressed very well. Nevertheless, I note that several recent VLM-related works on medical phrase grounding and explainable medical VQA are not included in the current comparison. I therefore suggest that the authors consider citing the following representative papers and clarifying their relationship to the proposed method: Medical phrase grounding with region-phrase context contrastive alignment Uncertainty-aware medical diagnostic phrase identification and grounding GEMeX: A large-scale, groundable, and explainable medical VQA benchmark for chest X-ray diagnosis

---

> > > ### Author Response · Authors · 2026-02-02
> > >
> > > We thank the reviewer for their positive assessment and for highlighting these relevant works. We will consider these references in the camera-ready version and clarify their relationship to our method.

---

### Official Review · Reviewer_iKNU · 2026-01-15

**Confidence:** 4
**Preliminary Rating:** 2
**Final Rating:** 2

**Summary:**

This paper addresses radiology report generation and visual grounding from 2D chest X-ray images using vision-language models (VLMs) enhanced with reinforcement learning (RL). The authors identify a key limitation of existing medical VLMs, namely their predominant reliance on supervised fine-tuning (SFT) without leveraging RL-based optimization. To address this, “RadVLM-GRPO” is introduced, a VLM framework that jointly improves report generation and visual grounding through three main components: (1) large-scale supervised training on chest X-ray data built on Qwen-3, (2) a cold-start SFT stage designed to incorporate explicit intermediate reasoning, and (3) a Group Relative Policy Optimization (GRPO) scheme that jointly rewards report generation quality and visual grounding clinical accuracy. Experimental results indicate that the proposed framework with the integration of RL leads to consistent performance improvements across both tasks.

**Strengths:**

1. The paper clearly motivates the use of reinforcement learning for radiology report generation and visual grounding by identifying important gaps in prior work, including: (i) the design of reward functions that account for both lexical quality and clinical correctness, (ii) a lack of systematic analysis disentangling the effects of RL objectives from intermediate reasoning strategies, and (iii) the necessity of domain-specific supervised fine-tuning when RL is applied within the same medical domain.

2. The authors provide a well-contextualized discussion of related work, covering RL, GRPO, and intermediate reasoning across large language models (LLMs), vision–language models (VLMs), and radiology-specific VLMs. This positioning helps clarify the limitations of existing radiology VLMs and motivates the individual components of the proposed method.

3. The experimental evaluation is employes a wide range of metrics for both report generation and visual grounding. Report generation is assessed using standard natural language generation metrics as well as clinically oriented efficacy metrics, while visual grounding performance is evaluated across anatomy, abnormality, and phrase grounding tasks.

4. An ablation study analyzes the contribution of each component of the proposed framework, demonstrating that the inclusion of online RL leads to consistent improvements in both report generation and visual grounding performance.

**Weaknesses:**

1. The proposed training pipeline involves multiple sequential stages, which may limit its practical applicability due to increased implementation complexity and computational overhead. Moreover, the method builds upon several existing training strategies and modules, which could be perceived as incremental and may reduce the perceived level of methodological novelty.

2. Despite the availability of implementation details and pretrained model weights, the paper lacks formalism in its presentation. In particular, the absence of mathematical formulations or explicit objective definitions makes the method harder to follow and reproduce, especially for readers seeking a clear understanding of the optimization procedure.

3. The experimental evaluation does not include k-fold cross-validation or repeated runs. While this omission may be justified by the high computational cost, it limits the statistical rigor of the quantitative analysis, as results are reported without standard deviations or statistical significance testing.

4. The paper does not present qualitative results within the main body (limited to 10 pages), which restricts deeper insight into model behaviour. Qualitative examples of generated reports and grounding visualizations would strengthen the empirical analysis and help assess clinical plausibility and failure modes.

**Detailed Comments:**

1. The paper would benefit from a clearer and more cohesive presentation of the quantitative results, particularly in Tables 1, 2, and 3. In the current layout, tables are not closely aligned with the corresponding discussion in the text, requiring frequent scrolling and making it harder to interpret the results. Placing each table nearer to its relevant paragraph or improving cross-referencing would substantially enhance readability.

2. Given the paper’s dual focus on report generation and visual grounding, the experimental section would be strengthened by the inclusion of qualitative results. Representative examples of generated reports and corresponding grounding or segmentation visualizations would provide valuable insights into model behaviour, complement the quantitative metrics, and help assess clinical relevance and typical failure cases.

**Justification Of Final Rating:**

I thank the authors for their detailed responses and for the clarifications incorporated into the manuscript.

This work addresses radiology report generation and visual grounding from 2D chest X-ray images using reinforcement learning. It presents a strong empirical study, including (i) the integration of RL for both report generation and visual grounding, (ii) an analysis of the respective impact of SFT, RL, and explicit reasoning, and (iii) the introduction of a continuous soft-F1 reward for visual grounding.

While these insights provide useful empirical evidence, and the release of code and model weights represents a valuable contribution to the community, the overall work appears closer to a carefully engineered, large-scale empirical study and a domain-specific foundation model effort than to a novel methodological contribution in machine learning for medical imaging.

**Justification Of The Preliminary Rating:**

This work addresses radiology report generation and visual grounding from 2D chest X-ray images through the integration of online reinforcement learning and intermediate reasoning training strategies. While the proposed approach demonstrates consistent improvements over strong baselines, the contribution appears closer in nature to a technical or systems-oriented advancement toward a domain-specific foundation model rather than a fundamentally novel methodological contribution. The framework relies on extensive pre-training at scale (e.g., 64 GPUs), builds upon an existing architecture and training pipeline (RadVLM), and incorporates incremental extensions such as explicit reasoning supervision and GRPO-based optimization.

The experimental evaluation is thorough in terms of the number and diversity of baselines considered across both tasks, and is further supported by an ablation study that helps disentangle the impact of individual components. However, the empirical analysis would be strengthened by the inclusion of qualitative results illustrating model behaviour in practice (some results are already present in supplementary materials). In addition, the methodology section would benefit from more rigorous formalism (e.g., explicit objective definitions), which would improve clarity, readability, and reproducibility.

Overall, the work represents a well-executed contribution toward large-scale radiology vision-language models, with clear empirical gains on most metrics, but offers more limited novelty at the level of core methodological innovation.

**Questions To Address In The Rebuttal:**

1. Contributions: Could the authors clearly enumerate their main methodological contributions and explicitly clarify how each component differentiates their approach from prior work in radiology VLMs and RL-based optimization?

2. Inference Efficiency: What is the average inference time per sample (for report generation and visual grounding), and how does it compare to relevant baselines in terms of computational efficiency?

---

> ### Author Response · Authors · 2026-01-24
>
> We thank the reviewer for the constructive feedback. In the following, we provide a detailed response and reference line numbers from the revised manuscript (new content highlighted in cyan).
>
> **Methodological Contributions and Differentiation from Prior Work**
>
> We have updated the manuscript to explicitly enumerate our methodological contributions (lines 81-91) and clarify how they differ from existing work:
>
> * **Methodological Contributions:**
>   * We systematically evaluate reasoning capabilities, finding that RL-optimized direct answers match or outperform thinking variants under our tested strategies (also discussed at l. 342-346).
>   * We introduce a novel continuous soft-F1 reward for visual grounding (highlighted l. 211-213) and benchmark multiple report generation rewards (Table 3).
>   * We release the full codebase and hyperparameters (SFT, rewards, GRPO) to facilitate full reproduction.
>   * Through ablation studies, we reveal the impact of different components (SFT, RL, thinking) on model performance.
>   * We analyze large-scale post-training RL for report generation and visual grounding.
> * **Differentiation from Prior Work:**
>   * Unlike DeepMedix-R1, which only focuses on reasoning traces, we isolate the impact of thinking by comparing against a GRPO-optimized non-thinking baseline. Furthermore, we treat visual grounding as a distinct downstream task with a reward, whereas they incorporate bounding boxes as intermediate steps, encouraged by the reward; the correctness of the bounding boxes within thinking traces does not influence the reward (discussed l. 134-140).
>   * CheXalign and CheXPO rely on offline DPO with pre-constructed preference datasets and do not explore thinking capabilities. Our approach uses online RL (GRPO), allowing the model to generate and learn from its own rollouts dynamically (mentioned l. 132-134).
>   * The CheXagent model relies solely on SFT without RL integration (mentioned l. 123-128).
>
> **Inference Efficiency**
>
> We now mention that on one GH200 node with 4 GPUs, on average over both tasks, RadVLM+RL has a throughput of 44.2 samples/s and RadVLM-Thinking+RL of 11.2 samples/s (added l. 229-232). We list a more detailed breakdown of the throughput of our models (using vLLM) and some baselines (the code provided by the baselines does not use an optimized inference library such as vLLM and is thus slower) in Appendix F:
>
> * Report Generation (vLLM): RadVLM+RL (45.7 samples/s), RadVLM-Thinking+RL (11.3 samples/s), CheXagent-2 (1.08 samples/s), DeepMedix-R1 (0.12 samples/s original code and 20.4 samples/s with vLLM).
> * Visual Grounding: RadVLM+RL (42.7 samples/s), RadVLM-Thinking+RL: (11.2 samples/s)
>
> **Pipeline Complexity and Novelty**
>
> We agree that multi-stage training pipelines increase complexity. (SFT, cold-start, RL). We would like to highlight the first RadVLM work that solely focused on the SFT stage (Deperrois et al., 2025) and its training analysis; this is why the current paper has a major focus on GRPO/RL training.
>
> Moreover, while we agree that our manuscript focuses more on the applicability of GRPO on CXR tasks rather than proposing a new RL optimization method specific to radiology, we believe that this task alone is complex and not thoroughly addressed in the current literature (SFT vs RL for CXR VLMs, GRPO training analysis, thinking vs non-thinking). We believe that our large-scale RL experiments and code base provide valuable insights into the current knowledge of RL in the CXR VLM domain (l. 81-91.).
>
> **Mathematical Formalism**
>
> We added mathematical formulations by moving the preliminary section from the appendix to the main text at lines 144-158, while keeping it concise to fit the manuscript length.
>
> **Statistical Rigor and Cross-Validation**
>
> We acknowledge limits to statistical rigor. As the reviewer suggested, we did not perform k-fold cross-validation or repeated runs due to the computational cost. One single run takes between 350-1000 GPU-hours (GH200) depending on the model (mentioned l. 225-226). Further, we would like to highlight that most baseline models only provide one trained checkpoint, making it difficult to capture potential variations in their re-run versions as compared to ours.
>
> However, we notice that RL consistently improves from SFT when we use different report generation rewards (except for RadGraph F1) in Table 3 (which are separate runs).
>
> **Qualitative Examples**
>
> We add a new Figure 3 in the main corpus containing: a) 3 comparison examples between RadVLM and RadVLM+RL for report generation, and b) 9 CXR images with predicted bounding boxes from all non-thinking ablated models. This addition brings more insight into how the model improves with RL at the level of generated report content and geometric overlap with reference boxes (referenced l. 277-282, 291-293). Example outputs of RadVLM-Thinking+RL are in Appendix K.
>
> **Presentation / Layout**
>
> We will try our best to improve it for the camera-ready version.

---

> ### Author Response · Authors · 2026-02-02
>
> We thank the reviewer for the constructive feedback, which helped us to improve the manuscript.
>
> While the final rating remains unchanged, we are encouraged that the reviewer explicitly recognizes our work as a "strong empirical study" and that it provides a "valuable contribution to the community".

---

### Author Rebuttal · Authors · 2026-01-24

**Rebuttal:**

We thank the reviewers for their constructive feedback and we have incorporated it into our revised manuscript.

The revised manuscript highlights new additions in cyan, deletions in red, and new figures in a darker blue.

We provide an overview of the changes:
* The title was changed to reflect the focus on chest X-Rays.
* Our contributions are now listed in the introduction section.
* We made changes to better highlight differences from previous work.
* We added a short preliminary subsection to the methods section, including mathematical formulas for SFT and GRPO.
* The distribution between tasks in the RadVLM SFT and cold-start datasets is now listed.
* We now mention GPU-hours required for SFT and GRPO training runs.
* Model throughput is now reported.
* We acknowledge the lack of human evaluation.
* A new Figure 3 contains qualitative examples of report generation and visual grounding of non-thinking models. There are now examples of outputs for thinking models, including the thinking process, in the appendix.
* We expanded the discussion section.

**Supporting Material:**

/attachment/cf4e04b9a76e8143c7836bdf3816431876328ea5.pdf

---

### Comment · Area_Chair_uqZe · 2026-01-31
**Final rating**

Dear reviewer, thank you for reviewing the paper! Please consider authors’ responses and update your final rating by clicking “Edit” → “Official Review” and providing the Final Rating by February 1st 2026 (23:59 AoE)!

---

> ### Comment · Area_Chair_uqZe · 2026-02-01
> **Final rating**
>
> Dear reviewer, thank you for reviewing the paper! Please consider authors’ responses and update your final rating by clicking “Edit” → “Official Review” and providing the Final Rating by February 1st 2026 (23:59 AoE)!

---

### Meta-Review · Area_Chair_uqZe · 2026-02-09

**Recommendation:** Accept (Poster)
**Confidence:** 4

**Metareview:**

Two reviewers recommend acceptance and highlight the experimental validation of the manuscript. While one reviewer recommends a weak reject (e.g., limited methodological contribution),  the reviewer also points out it is a strong empirical study.  Based on the reviewers’ feedback, I recommend acceptance of the paper.

---

### Decision · Program_Chairs · 2026-02-13

Accept (Poster)